# Preliminary Analysis of Long-Term Performance of a Piping: Aging and Creep Effects

**DOI:** 10.3390/ma14071703

**Published:** 2021-03-30

**Authors:** Salvatore Angelo Cancemi, Rosa Lo Frano

**Affiliations:** Department of Industrial and Civil Engineering (DICI), University of Pisa, 56126 Pisa, Italy

**Keywords:** long term operation, piping performance, aging, creep, thinning, corrosion, nuclear power plant, finite element analysis

## Abstract

Combining global experience, comprehensive aging knowledge, and predictive methodologies provides ideal prerequisites for the long-term operation strategy (LTO) of a nuclear power plant (NPP). Applying management strategies with an understanding of the ways in which structures relevant for the plant safety perform and interact in their operating environments is of meaningful importance for operating the plant beyond its originally licensed service life. In performing aging studies on the nuclear systems, structure, and components (SSCs), the results are crucial for demonstrating the safety and reliability of the NPP beyond 30 years of nominal operation. In this study, the synergistic effect of a creep mechanism with the alteration suffered by piping material is analyzed by means of MSC©MARC finite element code. Nonlinear analyses were performed to calculate the effects of the long operational period on a primary pipe, assess its degradation, and determine its residual functionality. In these analyses, both homogeneous and inhomogeneous pipe wall thinning are considered, as well as the operating or expected thermal–mechanical loads. The obtained results indicate that thermo–mechanical loads are responsible for pipe deformation, which develops and increases as the transient progresses. Furthermore, an excessive (general or local) wall thinning may determine a dimensional change of the pipe, even causing bending or buckling.

## 1. Introduction

Primary coolant systems in NPPs (nuclear power plants), such as piping, valves, piping elbows, etc., are exposed to damage mechanisms, which cause material characteristic deterioration as a consequence of the exposure to mechanical, physical, and chemical attacks during service, including high temperatures (from 270 to about 345 °C), high pressure (~15.5 MPa), corrosive coolant chemistry (e.g., change in pH), and radiation. SSCs (systems, structure, and components) thus suffer, aging during plant operation.

The procedure for testing suitability in an LTO (long-term operation strategy) involves a detailed screening of SSCs for aging evaluation to verify if they are capable of ensuring an acceptable residual safety margin for the life extension. As defined in [1], aging is the continuous time dependent deterioration of materials due to normal service conditions, which include normal operation and transient conditions; postulated accident and post-accident conditions are excluded [2,3,4,5].

Figure 1 depicts the number of operational nuclear power plants worldwide by mean lifespan as of July 2020 [6].

46% of reactors in operation have a lifetime between 31 and 40 years, while 19% are in operation for more than 40 years. Ensuring the long-term service of today’s nuclear power plants is a pressing policy and scientific issue. LTO increases the value of nuclear reactor assets.

Aging effects may be positive or negative. An example of a positive effect is reduced vibration from wear-in of rotating machinery, while negative effects are, e.g., the reduction in pipe diameter, cracking, thinning, or loss in material strength from fatigue or thermal aging.

Aging must be handled in order to ensure that especially the safety-related SSCs continue to guarantee their normal operation while being exposed to “damaging” conditions such as temperature, load, and environment to which the materials are exposed. This means that the safety level of each plant component and structure must always be maintained higher than its reference level [7,8]. Periodic inspections of the SSCs revealed unforeseen forms of degradation. These degradation phenomena occurred faster than anticipated, despite SSCs being designed against aging deterioration [9].

The objective of this study is to assess the performance of the 2nd generation pressurized water reactor (PWR) primary pipeline (class 1 of safety). Class 1 components are also the most difficult and complex to deal with because the physical conditions and the types of alteration and degradation phenomena (that were unthought-of of during the design stage) they face during the plant’s lifetime.

Particularly, the study focuses on PWR’s bent pipelines subjected to aging mechanisms such as thinning, thermal degradation, and creep effect. The impact of creep, together with the decrease of strength and the progress of aging, may impair the bearing capacity of the bent pipeline.

For this purpose, 3D thermo-mechanical finite element analysis was carried out to evaluate the effects and consequences possibly caused by the synergistic combination of aging, creep, and thinning in compliance with NPP life extension (beyond 30 years of operation). Among aging phenomena, we considered the thinning (homogeneous or localized-heterogeneous) caused by corrosion, which proceeds over the surface area of the material exposed, leading to a progressive reduction in the pipe thickness (a few tens of μm per year). If pipe thickness reduces too much, the pipe may collapse (buckling problem) under internal pressure [10]. Section 4 presents the results of the analysis; they are examined and discussed according to ASME III sect. NB-3232 [11].

It is noteworthy that very few studies investigating the effects/consequences of aging, creep, and thinning on the performance of SSCs in LTO are available in the open literature, as the attention of researchers, NPP owners, and regulators has been directed mainly, since the beginning of the nuclear era, on the issues of reactor pressure vessel (RPV) embrittlement.

Practical experience from operated plant and laboratory studies demonstrated that the thermal aging significantly affects the mechanical properties of duplex stainless steel materials and limit the reliability of the components. Fan et al. 2021 [12] highlight that the thermal aging embrittlement occurs after long-term service in the temperature range of 280–320 °C, leading to significant effects (degenerations in both the mechanical characteristics and corrosion resistance) on the microstructure of the stainless steel.

Cao et al. [13] observed that the microstructural evolution by spinodal decomposition occurred after 1000 h of aging, and that G-phase precipitation in the ferrite phase, after aging for 5000 h, was responsible for stainless steel plasticity reduction and for making it more sensitive to crack initiation. The same reduction in the material’s ductility was also observed by Silva et al. [14].

Shankar at el. [15] investigated the thermal aging of 316LN stainless steel at 1123 K for various time durations, showing that, beyond 10 h of aging (and up to 1000 h of aging), the material ductility, the yielding stress, and the total elongation decrease continuously because of the dislocation’s interaction with the microstructural constituents. Recently, interesting research by Liu et al. [16] studied experimentally the ratcheting behavior of a bending elbow pipe under different thermal aging periods of 1000 h and 2000 h at 500 °C. From the results, it emerged that the thermal aging period did not change the distributive law of ratcheting strains in the pipe’s middle cross-section; on the contrary, ratcheting strains were significantly different. Young’s modulus and the yield strength of austenitic steel were also strongly affected by thermal aging. In [16], a finite element model is proposed to predict the effect of thermal aging on nuclear SSCs.

In [17], Kim et al. investigated the effect of thermal aging embrittlement on the residual stresses of austenitic stainless steel repair welds in nuclear components subject to seismic loads. As a result, it was found that the effect of the thermal aging embrittlement on the residual stresses of the unrepaired welds is insignificant, whereas the tensile residual stresses of the repair welds increase in proportion to the increase in yield strength at room temperature. In addition, the residual stresses are significantly mitigated after the seismic loads. Fan et al. [18] studied the welding structures of primary coolant pipes subject to accelerated thermal aging. Mao et al. [19] investigated the RPV creep during a core meltdown scenario. They developed an advanced model for creep damage, capable of analyzing and predicting the fracture time and position within the RPV.

Hagihara and Miyazaki [20] performed a creep failure finite element analysis of piping subjected to local heating (for simulating severe accident conditions). The analytical results showed that the damage variable integrated into the creep constitutive equation was capable of predicting the pipe failure according to the test results performed by the Japan Atomic Energy Research Institute (JAERI), in which failure occurred from the outside of the pipe wall.

The studies available in the literature are limited when the object of the investigation is the simultaneous occurrence of aging, creep, and thinning phenomena that are expected during the extension of the life of the plant (beyond 30 years). It is thus crucial to understand if a plant, under such conditions, can still assure structural integrity. To that aim, in this study, aging, creep, and thinning were studied by means of finite element (FE) code in order to develop a methodology for thermo-mechanical performance prediction.

## 2. Creep Phenomenon

Generally, creep refers to the time-dependent plastic strain under constant load/stress at a given temperature, and it often becomes the life-limiting criterion for many SSCs. A thorough understanding of the plastic behavior of the material is so extremely important for the complete and rigorous design of SSCs.

Even through creep can occur at any temperature above absolute zero Kelvin, it is most commonly associated with time-dependent plastic strain at high temperatures, usually above 0.4 T_m_, in which T_m_ is the absolute melting point, since the diffusion may support creep at elevated temperatures [21]. It is widely recognized that high applied loads and temperatures may accelerate the rate of plastic deformation.

The strain rate–stress relationship, which typically exhibits three distinct regions: primary, secondary, and tertiary, is used to determine the creep behavior of materials (Figure 2). Furthermore, creep depends on material properties, stress, temperature, and time.

Figure 2 shows a typical creep curve with the three different regions: the primary, secondary, and tertiary creep regions:At an early stage/time, the creep resistance increases as the strain increases, resulting in reduced creep strain rate.In the secondary creep, the creep strain rate is almost constant (equilibrium between the work hardening and recovery processes).The strain rate increases exponentially up to the fracture in the tertiary creep; material undergoes deformation at a very high strain rate in less time up to failure. The tertiary creep is often regarded as “fracture mode”, and it is only observable at elevated temperatures and stresses.

There are several design methods utilized in the industry for analyzing creep problems [23]. In this study, we refer to a viscoplastic model expressed in terms of the following power law (called the Norton–Bailey power law) [22]:(1)$$d\varepsilon /dt=A\times \;\sigma^{n}\times k\times \;t^{k-1}$$
where:-A = coefficient,-*ε* = equivalent creep strain (–),-*σ* = equivalent stress (MPa),-*n* = stress dependence exponent,-*m* = temperature dependence exponent,-*t* = time (h),-*k* = time dependence exponent.

This model is suitable when primary and secondary creep dominate the history. The viscoplastic model, which was implemented in the FE analysis, has been verified and validated with reference to the experimental data of the FOREVER C/1 mock-up obtained from the melt vessel interaction (MVI) project [24]. The setting up of a 3D bent pipe model took place through the study of a multiaxial creep experiment and FE simulations [24]. The MVI was aimed at studying the risk of interactions induced by a core melt in a light water reactor during a severe accident, including creep. A 1:10 scale experimental rig was built to simulate the vessel behavior. The pressurization system was activated 20 min after the melt delivery. An inner pressure of 25 bar was kept within the vessel for 24 h, until the maximum creep strain reached 5%. Furthermore, FOREVER/C1 represents one of few experimental campaigns available in the open literature for performing multiaxial creep tests. Based on the measured experimental data, creep coefficients for power law equation are calculated through iterative fitting technique. These coefficients are verified and validated, and subsequently implemented in the 3D bent pipe model (Figure 3 and Figure 4).

The coefficients and exponents of Equation (1) are functions of the material properties. The coefficients are given in Table 1.

Furthermore, considering the few studies in the literature on the subject studied, it is believed that the smoothed mathematical function so obtained can, as a first approximation, represent the multiaxial behavior of the creep with a good approximation. For this reason, the application of the viscoplastic model to describe the combined effect of the creep, thinning, and aging is felt necessary.

## 3. Material and Methods

The pipeline performance (class I) was studied in compliance with the LTO program (beyond 30 years of operation). The combination of all three of the critical aspects, i.e., thermal aging, creep, and thinning effects, was taken into consideration in order to verify the structural integrity of such an item.

Compared to study [8], in which the aging and thinning effects are analyzed for a simple straight pipe, the behavior of an elbow pipe along its longitudinal direction is the subject of this study (complex geometry).

Thinning is more severe in bent pipe sections and elbows and in any component whose geometry varies drastically or is more complex; change in the direction of the flow makes an item susceptible to both erosion and corrosion damage [24]. The synergy between erosion and corrosion further aggravates the internal surface area damage.

For FE simulations, a 3D model representing a primary bent pipe made of 304L was set up and implemented. Figure 3 and Figure 4 show, respectively, the cross-section and the geometry of the bent pipe and the 3D meshed model. The part of the elbow pipe suffering thickness reduction is red colored in Figure 4. The model is completely symmetrical about the Z–Y plane without consideration of either the inlet/exhaust joint pipe or the flowing coolant.

The FE analysis aims to study the behavior of the item for 700 h beyond 30 years of service operation.

In the numerical assessment, we assumed thinning rates (W_r_) of the elbow equal to 0.5 mm/yr and 0.7 mm/yr, and 20% reduction of the nominal material properties [8,25]. Concerning corrosion/erosion phenomena, the thinning rate was determined based on the lesson learned from the Davis Besse NPP (Oak Harbor, OH, USA), the Japanese Mihama Unit 3 NPP accident [26], the study performed by Oh et al. [27], and the database [25].

The RPV vessel closure head inspections at the Davis Besse plant revealed a large cavity in the 15.24 cm thick low-alloy carbon steel RPV head material. This cavity was about 16.76 cm long and 10.16 to 12.70 cm at the widest point, extending down to the 0.635 cm thick type 308 stainless steel cladding. The 2004 accident in the Mihama Unit 3 NPP highlighted that the turbine return pipe, after 27 yrs in service, reduced its thickness to about 1.4 mm, corresponding to an 86% loss in thickness and too-harsh pressure–temperature conditions to sustain for the remaining pipe material.

The reduction of nominal steel material properties, procedures, correlations, and data for forecasting changes in mechanical properties of the stainless steel items are provided in [28]. Material data are calculated considering thermal aging (280–330 °C for 58,000 h) during service in light water reactors [29].

For FE analysis, after 30 yrs of nominal operation, the thickness of the wall is assumed at 35 mm and 29 mm for a thinning rate of 0.5 mm/yr and 0.7 mm/yr, respectively. These values are used as boundary conditions for the thermo-structural analysis. The reduction of pipe thickness is not updated throughout the transient period. To this end, the slight overestimation of residual life associated with the thickness variation of about 0.55 mm, calculated assuming W_r_ = 0.7 mm/yr for 700 h, does not affect the overall results.

The bent pipe is fully constrained at the two ends, i.e., clamps allowing expansion only in the Z and Y directions, respectively, for the long and short pipe run. 

With reference to the study [8], an inner pressure of 14 MPa and a constant temperature of 300 °C are set-up for the entire transient period of 700 h. Material properties of the numerical model are assumed to be temperature dependent, in accordance with Table 2, particularly those at 300 °C. The 304L properties are extracted from a materials database available in [25], collecting over 450,000 metal, polymer, ceramic, and composite material properties experimentally obtained in compliance with international regulations (e.g., ASME, ASTM). These validated data are hence implemented in the ageing model.

Von Mises criterion was chosen to measure the stress level.

A coupled thermo-mechanical viscoplastic analysis was performed in order to verify the structural integrity of the item. Multiple domains and independent or dependent variables representing various physical systems were included in the concept of coupled systems. When there were several domains involved, the solution for both domains was obtained at the same time.

The coupled systems can be divided into two categories:-Interface variables coupling: in this set of problems, a coupling happens via the domain interfaces. Domains can be physically different (e.g., fluid–solid interaction), or physically the same but with distinct discretization (e.g., mesh partition with explicit/implicit processes in different domains).-Field variables coupling: the domain may be the same or different. Coupling operates by differential equations that describe various physical phenomena, e.g., coupled thermo–mechanical problems.

Coupling between thermal and mechanical problems takes place by means of temperature-dependent material properties in the mechanical (stress) problem and internal heat generation in the mechanical problem induced by plastic work, which is used as input for the problem of heat transfer. Temperature distribution and displacement are then obtained. The effect of changes in the temperature distribution contributes to the deformation of the body through thermal strains and influences the material properties. There are two main reasons for the coupling. First, the coupling takes place when the deformation causes a shift in the related heat transfer problem, while the second cause of coupling is heat generated due to inelastic deformation. The irreversibility of plastic flow causes an increase in entropy in the body. The equations in matrix form for the thermal-mechanical problem are as follows:(2)$$Mü+D\dot{u}+K\left(T,u,t\right)u=F$$
(3)$$C^{T}\left(T\right)\dot{T}+K^{T}\left(T\right)=Q$$
where:-*M* = system mass matrix-*D* = damping matrix-*K* = stiffness matrix-*u* = nodal displacement-$\dot{u}$ = nodal velocity-$\ddot{u}\;$ = nodal acceleration-*T* = nodal temperature-*t* = time-*F* = nodal force vector-*C^T^* = heat capacity matrix-*K^T^* = thermal conductivity matrix-*Q* = heat flux vector

In Equations (2) and (3), the damping matrix *D*, stiffness matrix *K*, heat-capacity matrix *C^T^*, and thermal conductivity matrix *K^T^* are all dependent on temperature. The coupling between the heat transfer problem and the mechanical problem is as such due to the temperature-dependent mechanical properties and the internal heat generated. Thermo-structural analysis is a coupling analysis where the two physics passes are performed one after the other. The Kirchhoff constraints will tie the temperature of the bound node to the temperature of the projection point on its corresponding patch during the heat transfer pass of an increment.

The FE model is able to represent creep effect as the transient progresses by means of a viscoplastic model. The nodal force vector F in Equation (2) includes the contribution of various types of loading:(4)$$F=F_{\mathrm{point}}+F_{\mathrm{surface}}+F_{\mathrm{body}}+F^{*}$$
where F_point_ is the point load vector, F_surface_ is the surface load vector, F_body_ is the body (volumetric) load vector, and F^*^ represents all the other types of load vectors (for example, initial stress). Therefore, according to Equations (2)–(4), mechanical, thermal, and creep loads are updated automatically at each time step.

In the viscoplastic model, the plastic element is inactive for stress lesser than the yielding stress of the material [30]. As the transient progresses, the properties of 304L degrade, and the implemented creep model describes the primary and secondary stage.

Creep behavior is based on von Mises creep potential with isotropic behavior described by the power law seen in the previous Equation (1). The ratio thickness/diameter is less than 0.1, therefore the model is implemented with doubly curved thin shell elements. This selected element is a four-node, thin-shell type with global displacements and rotations as degrees of freedom. Bilinear interpolation is used for the coordinates, displacements, and the rotations. The element is defined geometrically by the (x, y, z) coordinates of the four corner nodes. Due to the bilinear interpolation, the surface forms a hyperbolic paraboloid, which is allowed to degenerate to a plate. The stress output is given in local orthogonal surface directions, V_1_, V_2_, and V_3_, which for the centroid are defined in Figure 5.

The shell elements are numerically integrated through the thickness, and the membrane strains are obtained from the displacement field. The number of layers through the thickness in the implemented model are eleven. The layer numbering convention is that the first layer is on the positive normal side of the shell, and the last layer is on the negative normal side.

The element normal is determined through the coordinates of the nodal position as well as the element’s connectivity. Simpson’s law is used to carry out the field variable integration for problems involving homogeneous materials [31].

## 4. Analysis Results

The performed coupled thermo-structural viscoplastic simulations focused on the study of the combined effects of creep, thinning, and aging for bent pipe beyond 30 years of nominal operation.

According to the study [8], five different cases have been considered:Case 1: item not affected by creep, thinning, and aging phenomena. This is the reference case.Case 2: 0.5 mm/yr with 10% of the steel properties’ reduction;Case 3: 0.5 mm/yr with 20% of the steel properties’ reduction;Case 4: 0.7 mm/yr with 10% of the steel properties’ reduction;Case 5: 0.7 mm/yr with 20% of the steel properties’ reduction.

The worst cases are represented by case 3 and case 5 for different W_r_.

A W_r_ of 0.5 mm/yr and 80% of steel properties should guarantee the structural integrity for 1.57 years beyond the 30 years of nominal plant operation, if W_r_ increases (e.g., up to 0.7 mm/yr) the structural integrity of the item is no longer guaranteed, as shown in Table 3. This means that the item must be replaced.

The results show that the most stressed part of pipeline is the outer elbow section, considering a W_r_ of 0.7 mm/yr, especially where the thickness is reduced. In Figure 6, Figure 7 and Figure 8, the top and bottom layers of equivalent von Mises stress and the total equivalent creep strain for the worst case, case 5, are represented.

Flexural deformation appears and becomes even worse when thinning increases. Eccentric bending may appear and determine cross-section ovalization, with or without bulging at the pipe’s outer surface. Additionally, it may be responsible for the buckling of the pipeline. Moreover, by analyzing the stress contour plot, it emerges that the flexural effects are not localized, but also extend to the straight part.

The results confirm the provision for a linear pipe in the study [8]; the structure reaches the plastic limit load after 121.45 h of nominal operation, as shown in Figure 9, Figure 10 and Figure 11. Instead, in case 3, the item does not reach the plastic limit load, and the thermo-mechanical analysis is completed as shown in Figure 12; the component may be considered adequate for the service.

The thinning, aging, and creep effects, as observed, strongly reduce the strength capacity, and their jeopardizing effects become even more relevant as long-term material corrosion, aging, and creep progress. This aspect is of fundamental importance in view of the life extension of existing NPPs, and for planning efficient plant life management.

## 5. Discussion

It was seen in this study that NPP operating environment and operating-history are critical aspects, together with time-dependent degradation phenomena, in determining whether plant SSCs may operate safely. The results of the thermo-mechanical analysis demonstrated that in case 2 and case 3, the pipe retains the required safety margin for long-term operation. Based on this analysis, the component is adequate for service operation. The thermo-structural viscoplastic model used demonstrated that is was a suitable tool for controlling the progression effects of thinning when aging and creep phenomena affect the structure; in this way, the residual life of SSCs can be determined.

The simultaneous effects of creep, thinning, and reduction of structural capacity due to aging jeopardize the structural integrity of the component, even with a small variation of the mechanical properties if W_r_ is equal to or greater than 0.7. In case 4, with 90% of nominal properties of AISI304, the component fails after about 272 h, while in case 5 (the worst), it fails after about 121.45 h. In case 5, with only a reduction of 10%, more of the steel’s thermomechanical properties in the item fail in half of the time (about 55%) when compared to case 4.

The localized thinning, aging, and creep effects, as observed, strongly reduce the strength capacity, and their consequences become even more relevant as life progresses [32]. The role of creep becomes marginal compared to thinning and aging even after 30 years of operation because of the given operating temperature (far lower than 0.4 T_m_).

The flow pattern in an elbow is subjected to great changes in direction and velocity, leading to significant differences in corrosion behavior at different locations (inhomogeneous thinning).

Equipment and material aging, degradation, and anomalies must be addressed in a thorough and timely manner in order to verify if the NPP item retains sufficient residual structural capacity and to guide corrective and mitigative actions, eventually.

## 6. Conclusions

In this paper, the performance of a pressurized elbow pipe subjected to the simultaneous action of creep and thinning phenomena was analyzed by means of a deterministic approach. For this purpose, an FE model was implemented to simulate the performance of an “aged” bent pipe beyond 30 years of nominal operation. Five different cases exemplifying the synergic effects of creep, thinning, and aging for the bent pipe were simulated. The results highlight:-In case 2 and case 3, the pipe retains a sufficient safety margin for LTO: the pipe is adequate for service operation.-For case 4 and case 5, the component fails at 272 h and 121.45 h, respectively, demonstrating that aging and a high thinning rate (0.7 mm/yr) threaten the pipe’s capacity (limit state is reached).-Flexural deformation is not localized, but also extends to the straight part of pipe. It becomes even worse as thinning increases.-Eccentric bending determines the ovalization of the pipe cross-section (buckling phenomenon).-The role of creep becomes marginal compared to thinning and aging even after 30 years of operation because of the given operating temperature (far lower than 0.4 T_m_).-Consequences of localized thinning, aging, and creep become even more relevant as the lifetime of the component progresses. The greater the materials’ degradation, the lower the residual resistance capacity and the greater the risk for the LTO.

Finally, the importance of the numerical modelling as a predictive and flexible tool capable of evaluating the performance of the degraded component (or similar components in parallel lines) and ensuring the safe operation of the plant was demonstrated.

## Figures and Tables

**Figure 1 materials-14-01703-f001:**
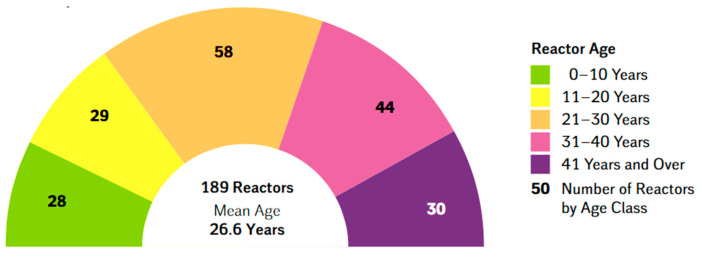
The overall age of the world’s nuclear reactors divided by the number of reactors currently operating (July 2020) [6].

**Figure 2 materials-14-01703-f002:**
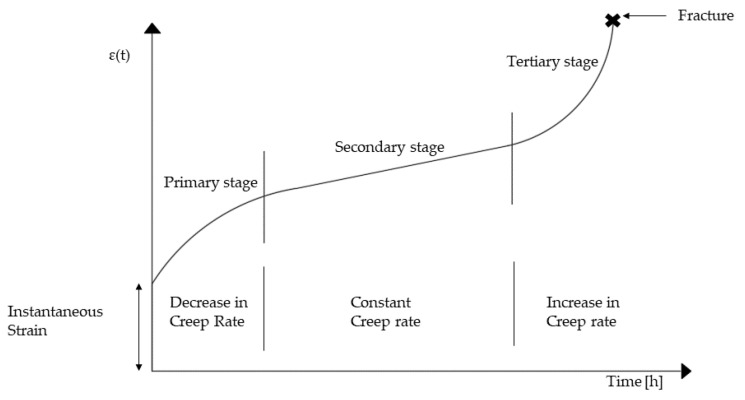
Creep curve under constant tensile load and constant temperature [22].

**Figure 3 materials-14-01703-f003:**
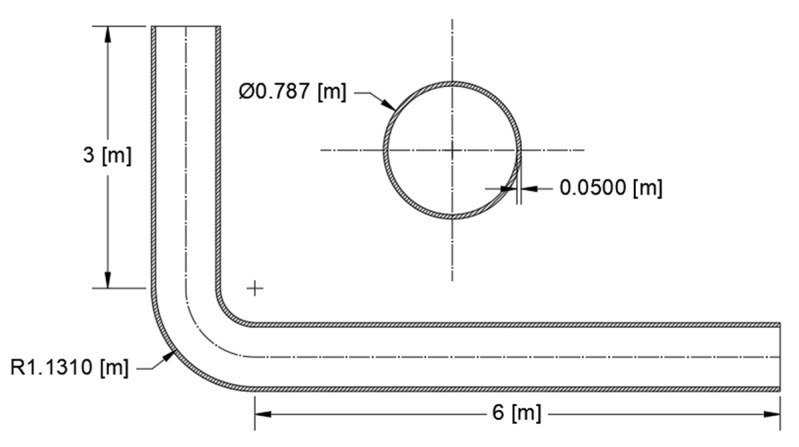
Cross-section of the bent pipe. Geometrical dimensions are in (m).

**Figure 4 materials-14-01703-f004:**
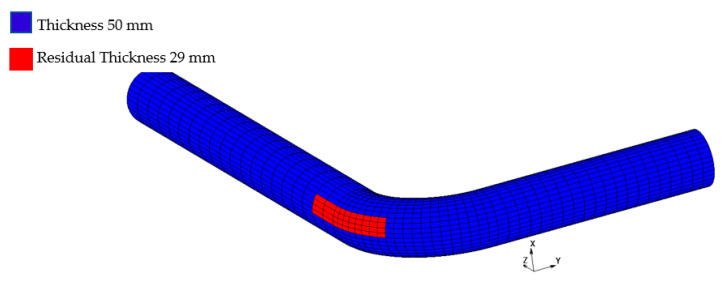
3D FE model of the bent pipe.

**Figure 5 materials-14-01703-f005:**
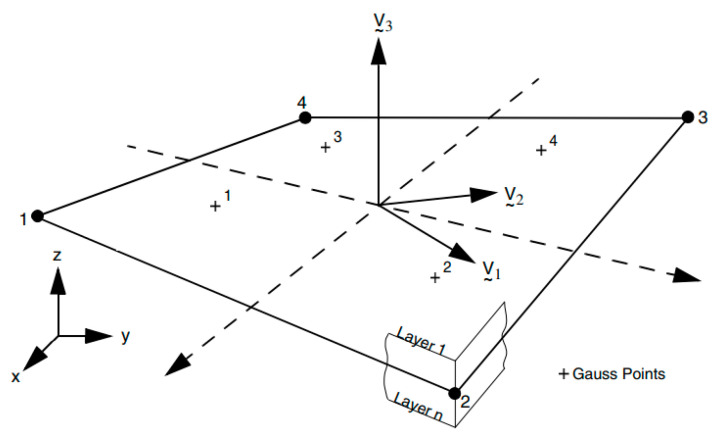
Element geometry basis.

**Figure 6 materials-14-01703-f006:**
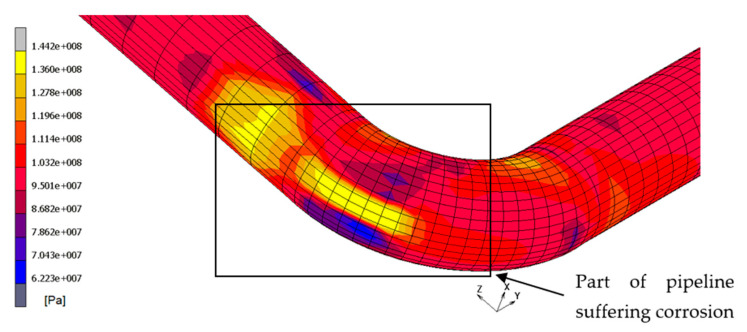
Equivalent von Mises stress, bottom layer—case 5.

**Figure 7 materials-14-01703-f007:**
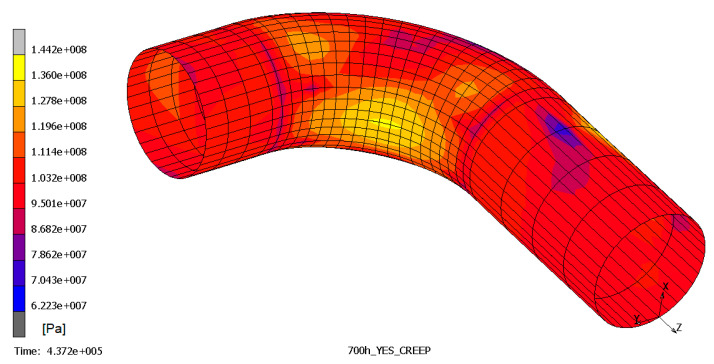
Equivalent von Mises stress, bottom layer—case 5.

**Figure 8 materials-14-01703-f008:**
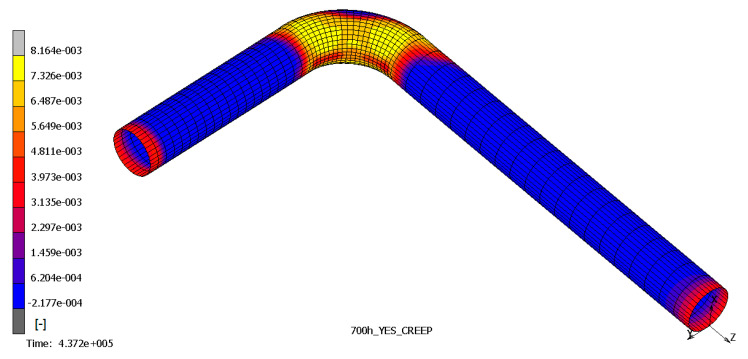
Total equivalent creep strain—case 5.

**Figure 9 materials-14-01703-f009:**
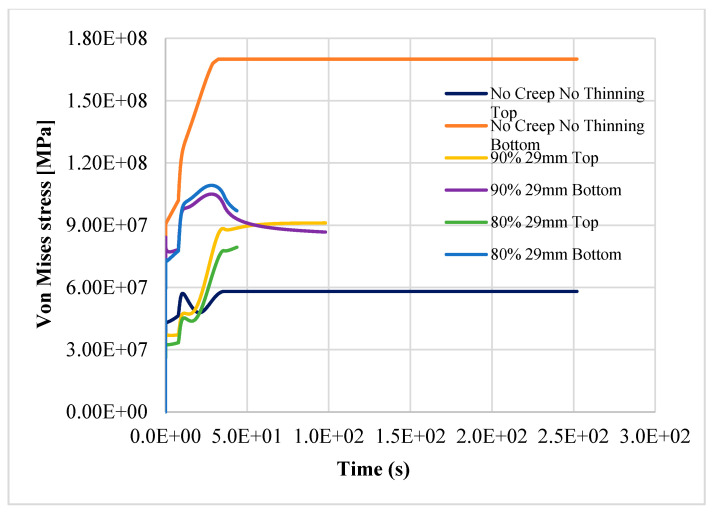
Top and bottom equivalent layers of equivalent von Mises stress away from thinning—case 1 (reference), case 4, case 5.

**Figure 10 materials-14-01703-f010:**
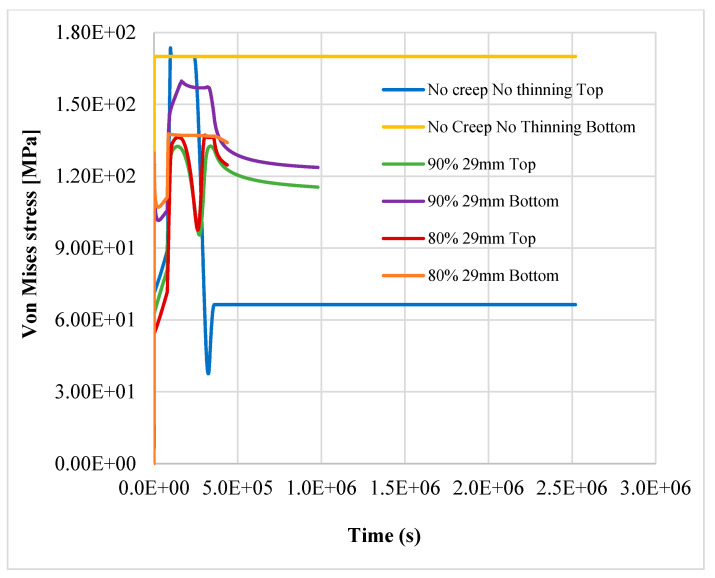
Top and bottom layers of equivalent von Mises stress in thinning—case 1 (reference), case 4, case 5.

**Figure 11 materials-14-01703-f011:**
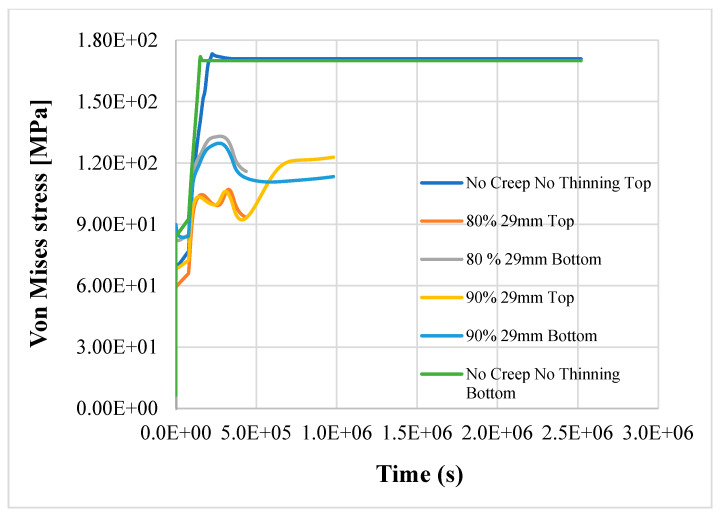
Top and bottom layers of Equivalent von Mises stress in thinning—case 1 (reference), case 4, case 5.

**Figure 12 materials-14-01703-f012:**
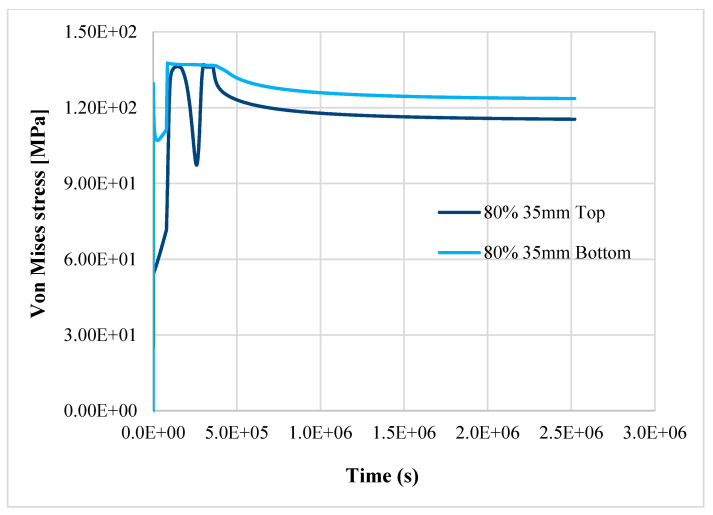
Top and bottom layers of equivalent von Mises stress in thinning—case 3.

**Table 1 materials-14-01703-t001:** Creep law coefficients.

A	n	k
1.64E-23	9.58	0.48

**Table 2 materials-14-01703-t002:** Thermo-mechanical properties of 304L [25].

Temperature (°C)	Density (kg/m^3^)	Specific Heat (J/kg K)	Conductivity (W/m K)	Thermal Expansion (×10^−5^ K^−1^)	Yield Stress (MPa)	Young’s Modulus (GPa)	Poisson’s Ratio (−)
−200	7900	157	8.4	1.22	412	181	0.294
−100	7900	3°0	12.6	1.43	319	181	0.294
0	7900	462	14.6	1.7	265	199	0.294
100	7880	496	15.1	1.74	218	193	0.295
200	7830	512	16.1	1.8	186	185	0.301
300	7790	525	17.9	1.86	170	176	0.31
400	7750	540	18	1.91	155	167	0.318
600	7660	577	20.8	1.96	149	159	0.326

**Table 3 materials-14-01703-t003:** Residual life (Lr) beyond 30 years of operation vs. 304L properties’ reduction [8].

Residual Life (y)		AISI304 Properties (% Nominal Value)
W_r_ = 0.5 mm/yr	W_r_ = 0.7 mm/yr	
15.72	2.66	100
9.46	–	90
1.57	–	80

## Data Availability

The data presented in this study are available on request from the corresponding author.

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
