# Peer review of "Preliminary Analysis of Long-Term Performance of a Piping: Aging and Creep Effects"

_materials, 2021, doi:10.3390/ma14071703_

Round 1

Reviewer 1 Report

Paper presents a study investigating the long-term performance of a piping material considering aging and creep effects analysed by means of MSC©MARC finite element code.

Manuscript is of appropriate quality, well written, clear and well organized. The results are correctly interpreted. Authors use suitable contemporary literature sources, and the text is supplemented with adequate illustrative figures and tables. Some minor inadequacies were only identified in the text. Thus, the paper is, in my opinion, suitable for publishing in Materials after minor revision.

List of inadequacies and recommendations to minor revision of text:

1) Carefully check the use and notation of all abbreviations in manuscript. For example, in Abstract the wrong abbreviation is written on line 13 (it is used SCCs instead of SSCs)

2) The line no. 122 – the marking is not in accordance with equation (1) on line 116

3) Different style is used in text for term “FE analysis” vs. “FE-analysis” – see lines 131, 154 etc. Please use the uniform style for the whole manuscript.

4) Figure 6 – its textbox is placed above the header of page 10. Please correct the position of figure according the rules of manuscript template.

5) References section - the list of references does not meet the manuscript template. Correct it carefully.

Author Response

All revisions are in .doc file. Thank you in advance for suggestions. 

Reviewer 2 Report

The paper subject is interesting application of modeling to nuclear power plants pipes simulation taking care about ageing. The authors use word “preliminary” in title of the paper and the presented research is preliminary and should be more deeply analysed. The authors published paper “Preliminary study of the effects of ageing on the long-term performance of NPP pipe” in 2021. The difference is more complex model (taking into account creep) in the paper. The numerical example is also for different geometry. The problem of aging pipes in nuclear power plants is serious and authors should publish not preliminary but advanced studies at least the numerical results should be validated using experimental setup.

The paper is a example of FEM calculations using MSC.Marc software. The presented approach is typical and can be used by most engineers trained in this software. Of course the topic is interesting due to the nuclear power plant case but nothing new from numerical modeling using FEM has been done.

The paper contains figures in poor resolution and some of them should be redrawn using computer.

Author Response

(The authors gave the same response as above.)

Reviewer 3 Report

In manuscript, the synergistic creep effect with the change in pipe materials was analyzed using the MSC © MARC finite element code. The paper is interesting, but requires significant improvements before publishing in Metals.

  1. Please modify and extend the entire discussion
  2. Lack of summary/conclusions
  3. In Figures 9, 10, 11, 12: describe the XY axis

Author Response

(The authors gave the same response as above.)

Round 2

Reviewer 2 Report

The paper was imporved but the improvemnts are not significant. No experimental verifications are shown so the presented results could be correct but also could be completly wrong.

Author Response

Thanks for suggestions

Reviewer 3 Report

The manuscript has been appropriately revised. Now, it is acceptable.

Author Response

We would like to thank the reviewer for his approval

This manuscript is a resubmission of an earlier submission. The following is a list of the peer review reports and author responses from that submission.

Round 1

Reviewer 1 Report

The article by Cancemi and Lo Frano deals with teh long term stability of representative material type intended for use in nuclear plant systems. The research is relevant and important and presents interestign insights in problems with part durability and its modeling. The methdos presented in the paper are adequate and appropriate to address the described research (and technological!) problem. Application of FEM is suitable approach. The presented research is important since it is difficult to model and predict the lifetime of materials under usage conditions for such extended periods. The manuscript may be suitable for publication after considering some major issues:

Language of the manuscript needs significant improvement. Some suggestions are (exemplarily!!) given below as more specific comments. Sometimes the text lacks precision (see specific comments below). Materials and methods section needs improvement, methods used are not sufficiently well described. Manuscript structure (subsections!) should be revisited since they do not follow the general suggestions made by the journal editorial office. I suggest usign the traditional structure of a scientific paper. Manuscript should be checked for missing data/links should be removed from the docment (I refer to page 5, last line in the paragraph: "Error! Reference source not found"; the same is with page 6 end of second paragraph. A brief discussion of the available literature should be included in the manuscript and the finding of calculations should be presented in light of current knowledge .

Authors should point out the benefit for other researchers a bit more clearly in the conclusions sections since it is not really clear inhowfar the state of the art has been extended by the author´s work. What´s the transferable part of what is presented? Please also drop some sentence why exactly the chosen geometry and the chosen parameters for bent pipe object was chosen for the study and comment on the generalizable aspects.

Specific comments

Abstract:

Line 1: "and methodologies" - of what?

Line 3: "necessary to deepen teh study" - improve the understanding

Line 4: "perturb" - impede

Line 6: "can be responsible of mechanical" - is responsible for

Line 7: "performances of a primary pipe of a typical Pressurized" - performance, pressurized.

Line 9: "have been" - were

Line 13: "[...] showed ... responsible of [...]" - show, responsible for

Introduction

Page 1

Line 1: "The LTO procedure" - In the abstract, LTO was defined as "long term operation". This definition does not fit with the sentence here, my suggestion is: "The procedure for testing suitability in LTO"

Line 3: delete "indeed", substitute "degradation" by "deterioration"

Line 8: "The 46 % [...] has [...] the 19% is" - please remove "the" in each case, exchange "has" against "have" and "is" against "are"

Page 2

Line 9: "ensure that its expected aging process ... anticipaated": please rephrase sentence for precision and clarity

Line 11: write a small "g" in "G"eneration

Line 18: "To the purpose" - use "To this end" or "For this purpose"; use comma

Line 19: "[...] caused possibly" -  possibly caused by

Line 20: "Synergic" - suggested is "synergistic"

Line 24: "section 4 presentS"

Line 26: "It is worthy to note" - it is noteworthy

Line 28: "beginning of THE nuclear area"

Line 29: "The studies become almost rare" - please rephrase

and so on, I stopped going through the language issues since there are by far too many. I suggest the authors go carefully through their manuscript improve teh anglish language significantly

Author Response

Please see the attachment .doc file. Regards.

Reviewer 2 Report

This paper presents a numerical analysis of a pressurized water reactor's long-term performance with a focus on creep-induced material degradation. The research topic is indeed worth investigating, yet the presented numerical workflow and the obtained results are not a novel contribution to the existing literature. From a reader’s perspective, I do not enjoy reading this manuscript as lots of the key information is missing.

Major comments:

  • The study focuses on the creep effect. Except for the section on creep phenomenon with the introduction of the power-law of creep strain rate, the authors failed to show whether the finite element analysis adopted the creep law or not? If it is, what are the parameters used for the power-law? Given the start of the numerical simulation is 30 years of regular operation, what is the creeping stage simulated?
  • The authors assumed a 0.5 mm/yr or 0.7 mm/yr of shell shining (corrosion) rate with a 20% reduction of nominal material properties, is there a reference for the numbers? For the corrosion rate, the authors should clearly state they are used to calculate the thickness of the target wall at the start of the simulation, and the corrosion is not numerically updated in the simulation. Also, does the reduction of the material properties corresponding to all the thermo-mechanical properties in Table 1?
  • The authors decoupled the corrosion and the creep using a pre-assumed corrosion rate in the first 30 years of regular operation and a creep analysis for the residual functional years. The omit of corrosion in the finite element analysis will over-estimate the residual life.

Minor comments:

Introduction:

  • The nonlinear finite element analysis is mainly due to the visco-plasticity, not the geometrical bending pip.
  • The logic of the introduction makes the reader confusing what the contribution of the manuscript is.
  • RPV is not explained in the first place.

Creep Phenomenon

  • What is the definition of the equivalent of the creep strain and the equivalent stress?
  • Does the coefficient of A in equation 1 have any relation with A in Figure 2?

Pipeline Performance Assessment

  • What is the mechanical constitutive model adopted for the simulation? Is the visco-plastic model of Eq.1 only used? If yes, how does the visco-plastic model coupled to the yield stress in Table 1?
  • The authors claimed 11 layers are used to analyze the pipe wall in the thickness direction. How are the layers coupled?

Analysis Results

  • The label in the y-axis of figures 8-11 should be listed.

Author Response

Please see the attachment .doc file. Regards

Round 2

Reviewer 1 Report

The authors have responded thoroughly and well to the raised issues. The manuscript is ok from my side

Reviewer 2 Report

The authors claimed their contribution is the coupled analysis of aging, creep, and thinning. Yet, They pre-assume a constant thinning rate with a constant properties reduction with no reference basis. The performed FE analysis decouples thinning, aging with creep, which does not reflect their claim.  

Also, the authors failed to address my major comments directly.